# Transcriptional Regulator TonEBP Mediates Oxidative Damages in Ischemic Kidney Injury

**DOI:** 10.3390/cells8101284

**Published:** 2019-10-20

**Authors:** Eun Jin Yoo, Sun Woo Lim, Hyun Je Kang, Hyun Park, Sora Yoon, Dougu Nam, Satoru Sanada, Mi Jin Kwon, Whaseon Lee-Kwon, Soo Youn Choi, Hyug Moo Kwon

**Affiliations:** 1School of Life Sciences, Ulsan National Institute of Science and Technology, Ulsan 44919, Korea; ejyoo89@unist.ac.kr (E.J.Y.); hjkang90@unist.ac.kr (H.J.K.); skyline@unist.ac.kr (H.P.); yoonsora1@unist.ac.kr (S.Y.); dougnam@unist.ac.kr (D.N.); wlee@unist.ac.kr (W.L.-K.); 2Transplantation Research Center, The Catholic University of Korea School of Medicine, Seoul 06591, Korea; swlim@catholic.ac.kr; 3Division of Nephrology, Japan Community Health Care Organization Sendai Hospital, Sendai, Miyagi 981-8501, Japan; satsanada@sendai-kidney.jp; 4PSA Co., Ltd., Nonsan-si, Chungcheongnam-do 32992, Korea; kwonmj1108@naver.com

**Keywords:** necrosis, apoptosis, acute kidney injury, reactive oxygen species, peroxisome, mitochondrial inner membrane

## Abstract

TonEBP (tonicity-responsive enhancer binding protein) is a transcriptional regulator whose expression is elevated in response to various forms of stress including hyperglycemia, inflammation, and hypoxia. Here we investigated the role of TonEBP in acute kidney injury (AKI) using a line of TonEBP haplo-deficient mice subjected to bilateral renal ischemia followed by reperfusion (I/R). In the TonEBP haplo-deficient animals, induction of TonEBP, oxidative stress, inflammation, cell death, and functional injury in the kidney in response to I/R were all reduced. Analyses of renal transcriptome revealed that genes in several cellular pathways including peroxisome and mitochondrial inner membrane were suppressed in response to I/R, and the suppression was relieved in the TonEBP deficiency. Production of reactive oxygen species (ROS) and the cellular injury was reproduced in a renal epithelial cell line in response to hypoxia, ATP depletion, or hydrogen peroxide. The knockdown of TonEBP reduced ROS production and cellular injury in correlation with increased expression of the suppressed genes. The cellular injury was also blocked by inhibitors of necrosis. These results demonstrate that ischemic insult suppresses many genes involved in cellular metabolism leading to local oxidative stress by way of TonEBP induction. Thus, TonEBP is a promising target to prevent AKI.

## 1. Introduction

Acute kidney injury (AKI) imposes a significant burden on the health care system as it affects more than 13 million people globally and is associated with increased morbidity, prolonged hospitalizations, and mortality [1,2,3]. Renal ischemia is a common cause of AKI along with sepsis and cisplatin treatment [4]. Kidney tubules are highly susceptible to ischemic injury and much of the cellular pathways involved have been delineated in recent years [5]. Ischemia causes acute tubular necrosis as well as apoptosis via several pathways of regulated necrosis: necroptosis which is caused by membrane rupture due to phosphorylation by the receptor-interacting protein kinase 3 [6], ferroptosis which is caused by failure of glutathione-peroxidase 4 function and lipid peroxidation due to endoplasmic reticulum stress and a lack of glutathione [7], and cyclophilin D-dependent mitochondrial-permeability transition-mediated regulated necrosis [8]. In addition, ischemia also causes local inflammation which is tightly linked to the necrosis: so called necroinflammation in which cellular injury and inflammation are reciprocally enhanced in an autoamplification loop [5]. Understanding molecular pathways leading to necroinflammation would be critical for developing therapeutic strategies for AKI.

Initially discovered as a DNA binding transcriptional enhancer in response to hypertonicity [9], TonEBP (tonicity-responsive enhancer binding protein) is a pleiotropic transcriptional regulator—both as a transcriptional cofactor [10,11] and a transcriptional suppressor [12,13,14,15]. In addition, TonEBP is a stress protein that is induced by a variety of stresses including hyperglycemia [16], excess calorie intake or hyperlipidemia [15], inflammation [10,17,18], and hypoxia [19]. Here we discover that TonEBP is induced by ischemia and mediates necroinflammation in the kidney by suppressing genes that encode proteins in cellular pathways of metabolism leading to local oxidative stress. Our data show that TonEBP is a critical mediator in AKI.

## 2. Materials and Methods

### 2.1. Animal Model

All the methods involving live mice were carried out in accordance with the approved guidelines. All experimental protocols were approved by the Institutional Animal Care and Use Committee of the Ulsan National Institute of Science and Technology. Five- to six-week-old heterozygous *TonEBP^+/Δ^* mice [20] that had been back-crossed for 10 generations onto the C57BL/6 background, as well as their wild-type littermates (WT, *TonEBP^+/+^*), were used. The mice were anesthetized with an intraperitoneal injectin of tiletamine (Zoletil, Virbac Laboratories, Carros, France) and xylazine hydrochloride (Rompun, Bayer, Leverkusen, Germany). Following an abdominal midline incision, both renal pedicles were bluntly dissected and clamped with a microvascular clamp (Roboz Surgical Instrument, Gaithersburg, MD, USA) for 30 min. During these procedures, mice were kept well hydrated with warm sterile saline at a constant temperature (37 °C). After the clamps were removed, the wounds were sutured and the mice were allowed to recover with free access to chow and water. Sham-operated mice were subjected to a similar surgical procedure except that the renal pedicles were not clamped. 24 h later (24 h reperfusion), mice were reanesthetized, blood was obtained by retro-obital bleeding, and kidneys were removed for analysis.

### 2.2. Immunoblot Analysis

The kidneys were perfused via heart with ice-cold PBS for 15 s to remove the blood. Cortex and outer medulla were rapidly dissected from each kidney on glass plates kept ice-cold. They were immediately homogenized for 20 s in hot (80 °C) lysis buffer (1% SDS, 1 mM sodium orthovanadate, 10 mM Tris-HCl, pH 7.5) using a polytran (Kinematica AG, Luzern, Switzerland) at full speed to minimize proteolytic degradation of TonEBP. The homogenates were cleared by centrifugation and separated on standard SDS gel electrophoresis for immunoblotting using antibodies against TonEBP at 1:3000 dilution [9], Bax at 1:1000 (Cell Signaling Technology, Danvers, MA, USA), Bcl-2 at 1:1,000 (Cell Signaling Technology, Danvers, MA, USA), and Hsc70 at 1:5000 (Rockland Immunochemicals Inc., Pottstown, PA, USA). The relative optical density of a band in each lane was normalized with the density of the Hsc70 band from the same gel.

### 2.3. RNA Extraction and Quantitative Real-Time PCR

Total RNA was extracted from kidney tissues and cells using TRIzol reagent (Invitrogen, Carlsbad, CA, USA). cDNA was synthesized with oligo d(T)_15_ and M-MLV reverse transcriptase (Promega, Madison, WI, USA). Quantitative PCR reactions were performed with SYBR Green I Master and LightCycler 480 II (Roche, Basle, Switzerland). The housekeeping gene Cyclophilin A (CypA) was used as a control to normalize the mRNA content of each sample.

### 2.4. Histopathology Examination

Kidneys were fixed in periodate-lysine-2% paraformaldehyde and embedded in paraffin, and 4-μm sections were cut and stained with Periodic acid-Schiff. Renal tubular damage was scored by a renal pathologist who was blinded to the experimental groups. A grade scale of 0–5, as outlined by Lien et al. [21], was used by counting the percent of tubules that displayed cell necrosis, loss of brush border, cast formation, and tubule dilatation as follows: 0 = none, 1 ≤ 10, 2 = 11–25, 3 = 26–45, 4 = 46–75, and 5 > 76%. At least 10 fields (original magnification × 100) were examined from each slide.

### 2.5. Immunohistochemistry

Immunohistochemistry was carried out as described previously [22]. Briefly, paraffin-embedded kidney 4-μm sections were deparaffinized with xylene and rehydrated in a graded alcohol series, then placed in citrate buffer solution (pH 6.0). Coplin jar containing slides were immersed in a Pyrex container and heated with microwaves for 15 min to enhance antigen retrieval. After cooling, sections were immersed in 3% hydrogen peroxide solution 30 min to block endogenous peroxidase, and then treated normal serum (DAKO, Carpinteria, CA, USA) for 30 min. The sections were then incubated with primary antibodies to 4-HNE (Abcam, Cambridge, UK), or 8-OHdG (JaiCA, Shizuoka, Japan) at 4 °C for overnight. The next day, the slides were incubated for 1 h at room temperature with the secondary antibody. After several washes, 3,3′-diaminobenzidine tetrahydrochloride was applied to the slides to develop brown color, then histologic analysis was performed on randomly selected fields using a color image analyser (TDI Scope EyeTM Version 3.0 for Windows, Olympus, Tokyo, Japan) by a pathologist blinded to the identity of the groups.

### 2.6. In Situ TdT-Mediated dUTP-biotin Nick End Labeling Assay (TUNEL) Assay

Apoptosis in tissue sections was visualized using ApopTag in situ apoptosis detection kit (Millipore, Billerica, MA, USA). TUNEL-positive cells were counted in 20 different fields in each section at 200× magnification.

### 2.7. Analyses of Blood and Urine

Creatinine and blood urea nitrogen (BUN) were measured from serum samples using colorimetric kits (Stanbio Laboratory, Boerne, TX, USA). Urine osmolality was measured by a vapor pressure osmometer (Vescor, Orlando, FL, USA). Sodium from the serum and urine was measured by flame photometry (Cole-Parmer, Niles, IL, USA). Fractional excretion of sodium was calculated using the following formula: FE_Na_ = 100 × (Na_urine_ × creatinine_plasma_)/ (Na_plasma_ × creatinine_urine_). Measurement of 8-hydroxy-2′-deoxyguanosine Oxidative DNA damage was evaluated based on the level of the DNA adduct 8-hydroxy-2′-deoxyguanosine (8-OHdG) in spot urine using a competitive enzyme-linked immunosorbent assay (Cell Biolabs, San Diego, CA, USA). The amount of urinary 8-OHdG level was normalized by the amount of creatinine.

### 2.8. HK-2 Cells

HK-2 cells (ATCC CRL-2190), a cell line derived from normal human kidney tissue, were cultured in DMEM supplemented with 10% (*v*/*v*) FBS. For cell death experiments, HK-2 cells were transfected with TonEBP targeted siRNA or scrambled siRNA using Lipofectamine 2000 (Invitrogen, Carlsbad, CA, USA) followed by hypoxia, ATP depletion, or hydrogen peroxide treatment. For hypoxic treatment, cells were placed in a hypoxia chamber (APM-30D, Astec Co., Fukuoka, Japan) for 24 h with a compact gas oxygen controller to maintain oxygen concentration at 1% by injecting a gas mixture of N_2_ and CO_2_. Control cells were incubated in a regular cell culture incubator with 21% O_2_. For ATP depletion, cells were treated with 10 μM of antimycin A (Sigma Aldrich, St. Louis, MO, USA), 10 mM of 2-deoxy-d-glucose (Sigma Aldrich, St. Louis, MO, USA), and 1 μM of ionomycin (Sigma Aldrich, St. Louis, MO, USA) for 3 h. For treatment with hydrogen peroxide, cells were incubated with 0.5 or 1 mM of hydrogen peroxide (H_2_O_2_, Sigma Aldrich, St. Louis, MO, USA) for 1 h. Cells were pretreated with 1 μM of cyclosporin A (Sigma Aldrich, St. Louis, MO, USA), 0.5 μM of ferrostatin-1 (Sigma Aldrich, St. Louis, MO, USA), or 5 μM of necrostatin-1 (Sigma Aldrich, St. Louis, MO, USA) for 1 h then exposed to ATP depletion or 1 mM of hydrogen peroxide for 1 h. For protein analysis, cells were transfected with TonEBP targeted siRNA or scrambled siRNA followed by hypoxia for 24 h, ATP depletion, or hydrogen peroxide treatment for 6 h. For RNA analysis, cells were treated for 3 h.

### 2.9. Cell Viability Assay

Cell viability was determined by measuring the reduction of 3-(4,5-Dimethyl-2-thiazolyl)-2,5-diphenyl-2*H*-tetrazolium bromide (MTT, Sigma Aldrich, St. Louis, MO, USA). To assess cell death, culture medium was collected at the end of treatment. The amount of lactate dehydrogenase (LDH) in culture medium was quantified using the LDH cytotoxicity detection kit (Clontech, Mountain View, CA, USA).

### 2.10. Annexin-V/propidium Iodide (PI) Double Staining Assay

Apoptotic cells were determined with an FITC-Annexin V Apoptosis Detection Kit (BD Biosciences, San Diego, CA, USA) according to the manufacturer’s protocol. Briefly, the cells were washed and incubated for 15 min at room temperature in the dark containing FITC-Annexin V and PI. Afterwards, apoptotic cells were analyzed by flow cytometry (BD Biosciences, San Diego, CA, USA).

### 2.11. Microarray Analysis

RNA samples were labeled and hybridized to SuperPrint G3 Mouse GE 8 × 60k Microarray kit (Agilent Technologies, Santa Clara, CA, USA). The raw intensity values were background corrected, log2 transformed and then quantile normalized. The microarray data were analyzed with gene set enrichment analysis. Gene set enrichment analysis for the KEGG pathway [23] and Gene Ontology [24,25] was performed using GSEA software [26]. Gene set size was limited by 10~300 and the gene-permuting method was used. The statistical significance between the two groups was assessed by false discovery rate (FDR) [27].

### 2.12. Visualization of ROS 

To detect reactive oxygen species (ROS), HK-2 cells grown on chamber slides (Lab-Tek, Nunc, Naperville, IL, USA) were treated for 3 h with hypoxia, ATP depletion, or hydrogen peroxide. After washing with PBS, the cells were treated for 30 min with 10 μM of 2′,7′-dichlorodihydrofluorescein diacetate (DCF-DA, Sigma Aldrich, St. Louis, MO, USA). After wash with PBS, DCF fluorescence image was obtained using an FV1000 confocal fluorescence microscope (Olympus, Tokyo, Japan).

### 2.13. Statistical Analysis

All data are expressed as means ± SEM. Statistical significance (*p* < 0.05) was estimated by an unpaired *t*-test for comparisons between two conditions. A one-way ANOVA was used for comparisons between more than two conditions. Tukey’s post hoc test was used for multiple comparisons. All statistics were performed with GraphPad Prism 8.2 software (GraphPad, San Diego, CA, USA).

## 3. Results

### 3.1. Renal TonEBP is Induced in Response to Ischemic Insult in Association with Inflammation and Tissue Injury

In hepatocytes, H_2_O_2_ stimulates TonEBP expression and H_2_O_2_-induced necrosis and inflammation are dependent on TonEBP [11]. Since TonEBP is also induced by hypoxia [19] and ischemic injury in the kidney is characterized by necroinflammation [5], we asked whether TonEBP played a role in the ischemic injury. Male B57BL/6 mice were subjected to a 30 min bilateral renal ischemia followed by reperfusion (I/R) for 24 h as described in Methods. We found that renal TonEBP expression increased in response to I/R both in the cortex and outer medulla (Figure 1). Since inflammation stimulates TonEBP expression and vice versa [10,17], we investigated whether there were changes in local inflammation. mRNA expression of inflammatory cytokines, chemokines, and adhesion molecules was robustly induced in response to I/R in the outer medullae (Table 1). Among them, IL-6, IL-1β, IL-10, and MCP-1 mRNA increased more than 10-fold. Expression of these genes was also elevated in the renal cortex but the increase was smaller (data not shown). Widespread necrosis was also observed mostly in tubular structures in association with an increase in lipid peroxidation detected by 4-hydroxynonenal, a marker of ferroptosis (Figure 2). In addition, there were clear signs of apoptosis based on TUNEL-positive cells and higher expression of Bax in combination with lower expression of Bcl-2 (Figure 3). Renal function was also compromised dramatically: serum creatinine, blood urea nitrogen (BUN), and fractional excretion of sodium escalated while urinary osmolality dropped (Figure 4). As expected, the expression of mRNA for kidney injury molecule-1 (KIM-1) increased prominently. Thus, renal TonEBP expression increased in response to I/R in correlation with local inflammation, and histological and functional tissue injury.

### 3.2. TonEBP Deficiency Protects Kidneys from Ischemia-Induced Inflammation and Tissue Injury

We asked whether the inflammation and renal injury in response to I/R were affected by TonEBP deficiency. A TonEBP haplo-deficient line of mice (*TonEBP^+/Δ^*) [20] on B57BL/6 background were obtained as littermates of the wild type animals discussed above (*TonEBP^+/+^*). TonEBP deficiency in the cortex and outer medulla was confirmed based on immunoblot analyses (Figure 1). While TonEBP expression increased in response to I/R in the *TonEBP^+/+^* animals, it did not increase in the *TonEBP^+/Δ^* animals. Among the inflammatory genes whose expression increased in response to I/R in the *TonEBP^+^**^/+^* animals, many of them including IL-6 and MCP-1 showed a significantly smaller increase in their expression in the *TonEBP^+/Δ^* animals (Table 1) as expected from TonEBP deficiency. These animals also displayed milder tubular necrosis and lipid peroxidation (Figure 2), fewer TUNEL-positive cells, lower expression of Bax and higher expression of Bcl-2 (Figure 3). The increase in serum creatinine, BUN, and fractional excretion of sodium were tempered along with improved urinary osmolality plus a reduced expression of KIM-1 mRNA (Figure 4). In sum, TonEBP haplo-deficient animals were protected from the I/R-induced renal inflammation and injury suggesting that TonEBP played a role.

### 3.3. TonEBP Mediates Renal Tubular Cell Death in Response to Ischemic Insult

Since tubular necrosis in response to I/R was significantly milder in the TonEBP haplo-deficient animals (Figure 2), we asked whether TonEBP was involved. We addressed this question using a human renal epithelial cell line, HK-2 cells. We found that HK-2 cells displayed cell death in response to hypoxia (24 h in 1% oxygen) as indicated by reduced cell viability and increased LDH release (Figure 5A). The cell death was also observed in response to ATP depletion and treatment with H_2_O_2_ in a dose-dependent manner. The cell death in response to ATP depletion and H_2_O_2_ was blocked by various inhibitors of necrosis—necrostatin-1, ferrostain-1, and cyclosporin A—confirming that at least three forms of necrosis, i.e., necroptosis, ferroptosis, and mitochondrial-permeability transition-mediated necrosis, respectively, contributed to the cell death (Figure 5B). Importantly, siRNA-mediated knockdown of TonEBP prevented cell death in all the conditions indicating that TonEBP mediated the necrotic cell death.

Renal ischemia also leads to apoptosis in association with elevated Bax expression and reduced Bcl-2 expression as shown in Figure 3. We examined apoptosis in HK-2 cells using annexin V externalization and staining with propidium iodide. Hypoxia, ATP depletion, or H_2_O_2_ induced apoptosis (Figure 6) in association with escalated Bax expression and lower Bcl-2 expression (Figure 7). Interestingly, TonEBP was significantly induced in response to hypoxia, ATP depletion, or H_2_O_2_ (Figure 7, top panel). TonEBP knockdown prevented apoptosis (Figure 6) in association with lower Bax expression and higher Bcl-2 expression (Figure 7) in all the conditions, demonstrating that TonEBP mediated the ischemia-induced apoptosis.

Taken together, the results in Figure 5, Figure 6 and Figure 7 demonstrate that in HK-2 cells 1) TonEBP is induced by ischemic insults and 2) tubular necrosis and apoptosis are prevented by TonEBP deficiency. These observations are consistent with the induction of TonEBP in response to I/R, and the reduced tubular necrosis and apoptosis in response to I/R in the TonEBP deficient animals described above. The induction of TonEBP in response to I/R insult may be critical to necrosis and apoptosis in the kidney. This idea was tested in the next section.

### 3.4. TonEBP Promotes Oxidative Stress via Suppression of Metabolic Genes

TonEBP regulates many genes using a variety of action mechanisms—DNA binding transcription factor [9], transcriptional co-factor [10,11], and epigenetic suppressor [12,13,14,15]. In order to understand how TonEBP mediated the tissue damage and inflammation in response to I/R, we analyzed changes in the renal transcriptome using microarray analysis as described in Methods. Bioinformatic analyses revealed that several pathways—peroxisome, mitochondrial inner membrane, PPAR signaling, and glycolysis/gluconeogenesis—were significantly elevated in the TonEBP deficient animals over their wild type littermates after I/R (Table 2 and Figure 8). Among 77 genes in the peroxisome pathway, 25 genes (33%) showed significantly elevated expression in the TonEBP deficient animals. Likewise, 86 out of 291 genes (30%) in the mitochondrial inner membrane, 21 out of 79 genes (27%) in the PPAR signaling, and 18 out of 56 genes (32%) in the glycolysis/gluconeogenesis pathway displayed elevated expression in the TonEBP deficient animals after I/R. While many other genes showed significant suppression in the TonEBP deficient animals after I/R, they did not aggregate significantly into any particular pathway. Thus, those genes suppressed by TonEBP after I/R might have functional consequences while other genes stimulated by TonEBP might not.

We noted that most of the elevated genes in the 4 pathways discussed above were suppressed after I/R in TonEBP sufficient (wild type) animals (Figure 8). 22 out of the 25 genes (88%) in the peroxisome pathway, 76 out of the 86 genes (88%) in the mitochondrial inner membrane pathway, 19 out of 21 genes (91%) in the PPAR signaling pathway, and 13 out of 18 genes (72%) in the glycolysis/gluconeogenesis pathway were significantly suppressed after I/R in TonEBP sufficient animals. We confirmed these changes in selected genes using quantitative RT-PCR of kidney samples (Figure 9, left): expression of Acox2 (Acyl-CoA Oxidase 2), Pecr (Peroxisomal Trans-2-Enoyl-CoA Reductase), Pck1 (Phosphoenolpyruvate Carboxykinase 1), and G6pc (Glucose-6-Phosphatase Catalytic Subunit) decreased significantly after I/R while expression of Hmgcs2 (3-Hydroxy-3-Methylglutaryl-CoA Synthase 2), Slc25a25 (Solute Carrier Family 25 Member 25), Cyp4a14 (Cytochrome p450, Family 4, Subfamily A, Polypeptide 14), and Pklr (Pyruvate Kinase L/R) did not. Taken together, the results in Figure 9 suggest that the 4 pathways are suppressed in response to I/R because of the induction of TonEBP by ischemic insults. In HK-2 cells, some of the genes in the 4 pathways such as Acox2, G6pc, and Pklr were suppressed in response to ischemic insults but other genes were not. On the other hand, most of them showed enhanced expression in response to TonEBP knockdown under ischemic conditions (Figure 9, right) consistent with the idea that TonEBP suppressed their expression.

Many of the genes in the family pathways encode enzymes involved in lipid and carbohydrate metabolism, and components of mitochondrial ATP synthesis including 8 genes for proton transporting ATP synthase, 13 genes for complex I and 6 genes for complex III. We suspected that suppression of these genes would lead to oxidative stress, or production of ROS, in the renal tubular cells. In order to test this idea, we examined 8-hydroxy-2′-deoxyguanosine (8-OHdG), a marker of oxidative stress, in the kidney. Immunohistochemial analysis showed that 8-OHdG was abundantly detected in the kidneys after I/R, and the signal was reduced in animals with TonEBP haplo-deficiency (Figure 10A). Urinary excretion of 8-OHdG also increased in response to I/R, and the excretion was significantly lower in TonEBP haplo-deficiency (Figure 10B). In HK-2 cells we examined ROS using the conversion of 2′,7′-dichlorofluorescin diacetate (DCF-DA) to fluorescent DCA (Figure 10C). DCF fluorescence was dramatically induced in response to hypoxia, ATP depletion, or H_2_O_2_ demonstrating that ischemic insult led to robust production of ROS. As expected, the increased fluorescence was blunted by TonEBP knockdown. These observations demonstrate that ischemic insult causes oxidative stress in a manner dependent on TonEBP, i.e., via suppression of metabolic genes.

## 4. Discussion

Due to high mitochondrial content and energy demand in combination with unique hemodynamic characteristics, kidney proximal tubules are readily susceptible to ischemic injury leading to AKI. In fact, AKI due to ischemia, sepsis, and cisplatin share common features: necrosis and apoptosis of the proximal tubules in association with pathological changes in mitochondria – reduced biogenesis, swelling and physical disruption with loss of mitochondrial respiratory proteins. The damaged mitochondria release ROS, cytochrome c and mitochondrial DNA leading to cellular injury due to apoptosis and necroinflammation [4]. The mitochondrial dysfunction and ROS production are critical because the restoration of mitochondrial biogenesis using chemical agents blocks cisplatin-induced AKI [28,29]. Likewise, the administration of antioxidants [30] and hyperactivation of anti-oxidative transcription [31] prevents AKI in response to ischemia. The data presented here uncover that renal TonEBP is induced in response to ischemic insult due to the ROS produced by mitochondrial damage (Figure 11). This is supported by the observations in HK-2 cells where TonEBP is induced by hypoxia and ATP depletion as well as H_2_O_2_, each of which stimulates ROS production. Since TonEBP is also induced by lipopolysaccharide [10,14] and cisplatin (unpublished), TonEBP is likely to mediate AKI induced by sepsis and cisplatin as well.

Another salient finding of this study is that the elevated TonEBP expression in response to ischemic insult is critical for local production of ROS or oxidative stress. Among the genes shown in Table 2 and Figure 8, suppression of 111 genes in the peroxisome and mitochondrial inner membranes are likely to contribute to the ROS production. This represents a “second wave” of ROS production as it depends on the induction of TonEBP in response to the initial ROS produced by mitochondrial damage (Figure 11). In other words, the TonEBP induction is a link for maintaining ROS production and kidney injury. This is supported by the milder kidney injury in TonEBP deficient mice and protection of HK-2 cells whose TonEBP is knocked down from ischemic insult. In hepatocytes, ROS also induces TonEBP which in turn promotes cellular injury [11]. TonEBP appears to be a general mediator of ROS-induced cellular injury.

TonEBP mediates chronic inflammation by way of controlling gene expression. In macrophages, TonEBP promotes M1 phenotype by stimulating the transcription of proinflammatory cytokines and enzymes [10,18]. In addition, TonEBP suppresses the M2 phenotype by transcriptional suppression of PPARγ, IL-10, and HO-1 [12,13,14]. Thus, in macrophages TonEBP simultaneously regulates two opposing groups of genes–stimulation of M1 genes and suppression of M2 genes. This study uncovers that TonEBP contributes to the ROS production in the renal tubules by suppression of genes involved in the function of peroxisome and mitochondrial inner membrane, and cellular pathways of PPAR signaling and carbohydrate metabolism. TonEBP appears to be a stress protein responding to inflammatory signals in macrophages, oxidative stress in hepatocytes and mitochondrial injury in renal proximal tubules. Genome wide association studies have linked single nucleotide variations within the introns of the TonEBP gene with inflammation and renal injury [16], and type 2 diabetes mellitus risk [32]. As such, targeting TonEBP might prove to be efficacious against preventing chronic inflammation and AKI. Since specific inhibitors of TonEBP are not known, knockdown of TonEBP using siRNA or TonEBP-targeting lentivirus could be used.

## Figures and Tables

**Figure 1 cells-08-01284-f001:**
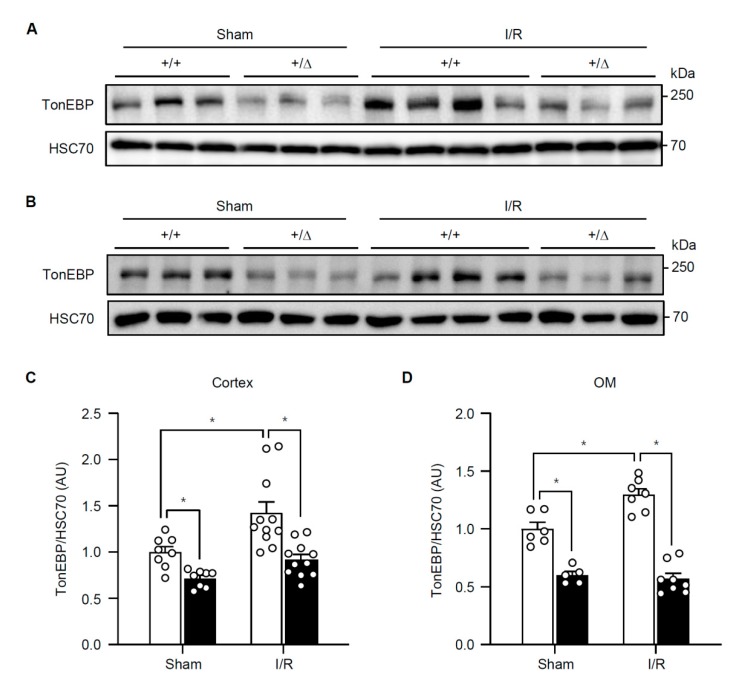
TonEBP (tonicity-responsive enhancer binding protein) expression in *TonEBP^+/Δ^* (+/Δ, filled bars) mice and their *TonEBP^+/+^* littermates (+/+, open bars) after ischemia/reperfusion (I/R) or sham treatment of kidneys. TonEBP and Hsc70 immunoblot were performed from renal cortices (**A**) and renal outer medullae (OM) (**B**), (**C**,**D**) Ratio of TonEBP and Hsc70 band intensity was determined and shown in arbitrary unit (AU). Mean + SEM, * *p* < 0.05.

**Figure 2 cells-08-01284-f002:**
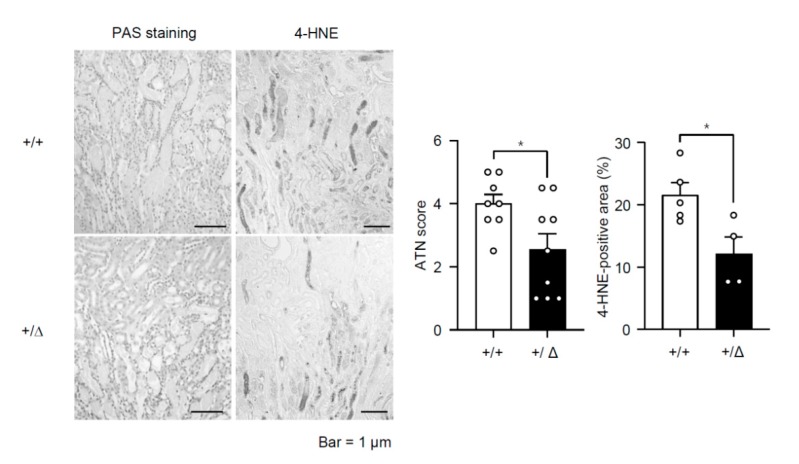
Renal tissues were obtained from *TonEBP^+/Δ^* (+/Δ, filled bars) mice and their *TonEBP^+/+^* littermates (+/+, open bars) after I/R treatment of kidneys. Tissue sections were stained with periodic acid-Schiff stain (PAS) and acute tubular necrosis (ATN) score was obtained. Tissue sections were also immunostained for 4-hydroxynonenal (4-HNE). 4-HNE positive area (%) was measured. Mean + SEM, * *p* < 0.05.

**Figure 3 cells-08-01284-f003:**
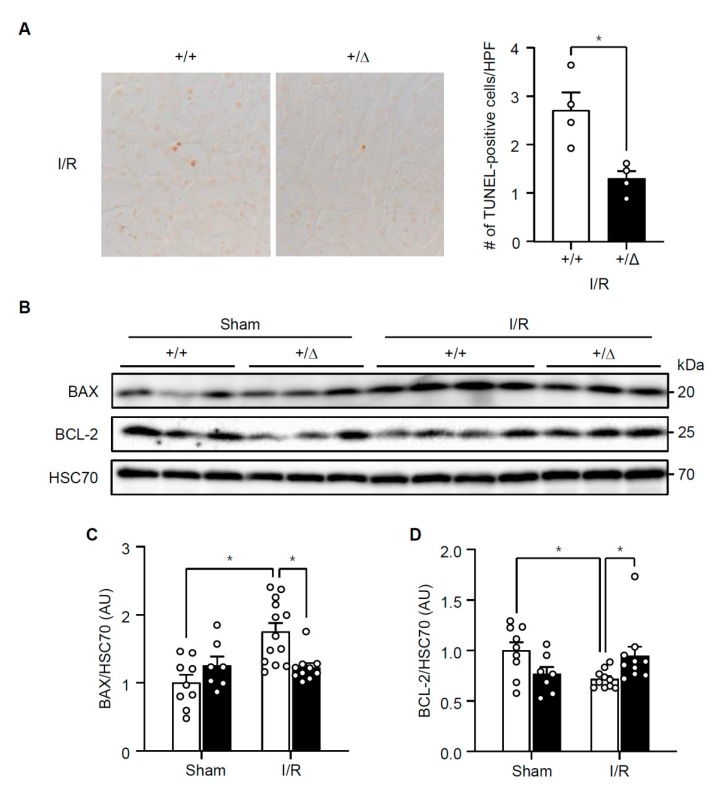
Renal apoptosis and expression of apoptotic proteins in *TonEBP^+/Δ^* (+/Δ, filled bars) mice and their *TonEBP^+/+^* littermates (+/+, open bars) after I/R or sham treatment of kidneys. (**A**) Kidney sections were stained for TUNEL. TUNEL-positive cells were counted and expressed as number per high power field (HPF), (**B**) Renal cortices were immunoblotted for Bax, Bcl-2, and Hsc70, (**C**,**D**) Ratio of band intensity, Bax/Hsc70, and Bcl-2/Hsc70, was calculated and shown in arbitrary unit (AU). Mean + SEM, * *p* < 0.05.

**Figure 4 cells-08-01284-f004:**
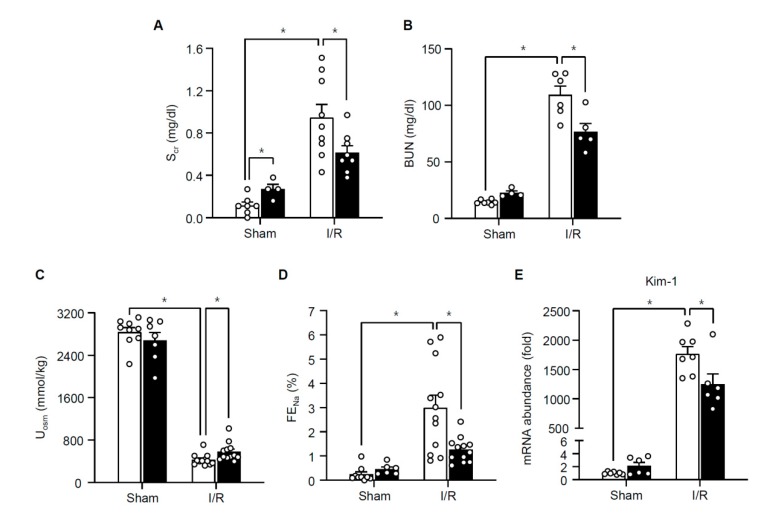
Serum creatinine (S_cr_, **A**), blood urea nitrogen (BUN, **B**), urine osmolality (U_osm_, **C**), fractional excretion of sodium (FE_Na_, **D**), and mRNA abundance for Kim-1 in renal cortices (**E**) from *TonEBP^+/Δ^* (filled bars) mice and their *TonEBP^+/+^* littermates (open bars) after I/R or sham treatment of kidneys. Mean + SEM, * *p* < 0.05.

**Figure 5 cells-08-01284-f005:**
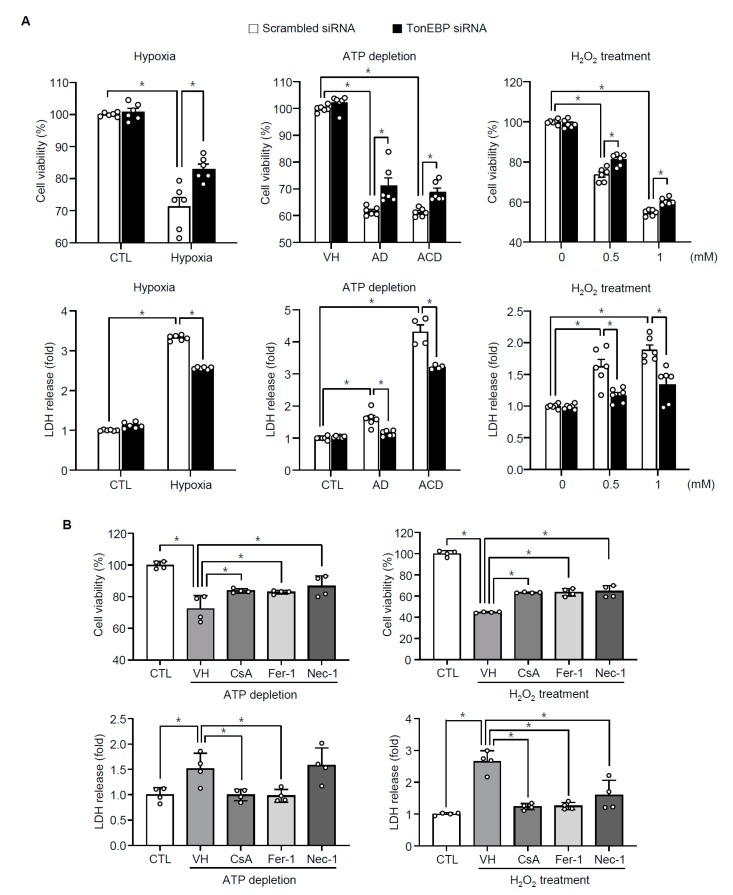
Effects of TonEBP knockdown and inhibitors on cell viability and release of lactate dehydrogenase (LDH). (**A**) HK-2 cells were transfected with TonEBP targeted siRNA (filled bars) or scrambled (non-targeting) siRNA (open bars). Cells were then incubated with hypoxia (1% O_2_) for 24 h, ATP depletion (AD—10 μM of antimycin A and 10 mM 2-deoxy-d-glucose, or ACD—10 μM of antimycin A, 10 mM 2-deoxy-d-glucose and 1 μM of ionomycin) for 3 h, or H_2_O_2_ treatment for 1 h. (**B**) Cells were pretreated for 60 min with vehicle (VH), cyclosporin A (CsA, 1 μM), ferrostatin-1 (Fer-1, 0.5 μM), or necrostatin-1 (Nec-1, 5 μM) followed by ATP depletion for 3 h or treatment with 1 mM H_2_O_2_ for 1 h. CTL, not treated. Mean + SEM, * *p* < 0.05.

**Figure 6 cells-08-01284-f006:**
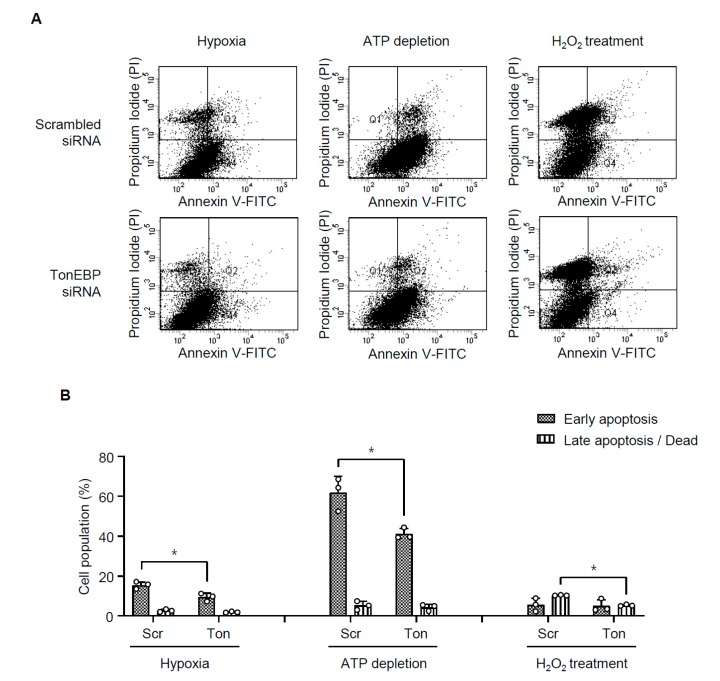
Effects of TonEBP knockdown on apoptosis. (**A**) HK-2 cells were transfected with TonEBP targeted siRNA or scrambled siRNA followed by hypoxia, ATP depletion, or H_2_O_2_ treatment. The cells were then labeled with propidium iodide (PI) and annexin V and analyzed using fluorescence-activated cell sorting. (**B**) Percentage of cells of early apoptosis (positive for annexin V and negative for PI) and late apoptosis / dead (positive for both PI and annexin V) is shown in Mean + SEM, * *p* < 0.05.

**Figure 7 cells-08-01284-f007:**
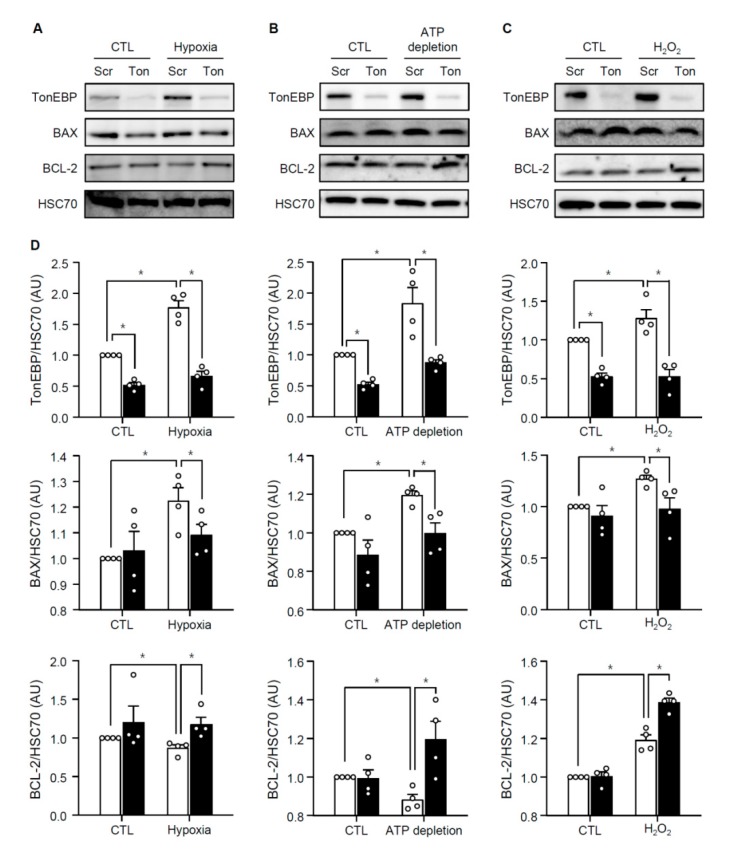
Effects of TonEBP knockdown on the expression of Bax and Bcl-2. HK-2 cells were transfected with TonEBP targeted siRNA (Ton, filled bars) or scrambled siRNA (Scr, open bars) followed by hypoxia (**A**), ATP depletion (**B**), or H_2_O_2_ treatment (**C**). CTL, not treated. Cell lysates were immunoblotted for TonEBP, Bax, Bcl-2 and Hsc7, (**D**) TonEBP/Hsc70, Bax/Hsc70 and Bcl-2/Hsc70 ratio are shown. Mean ± SEM, * *p* < 0.05.

**Figure 8 cells-08-01284-f008:**
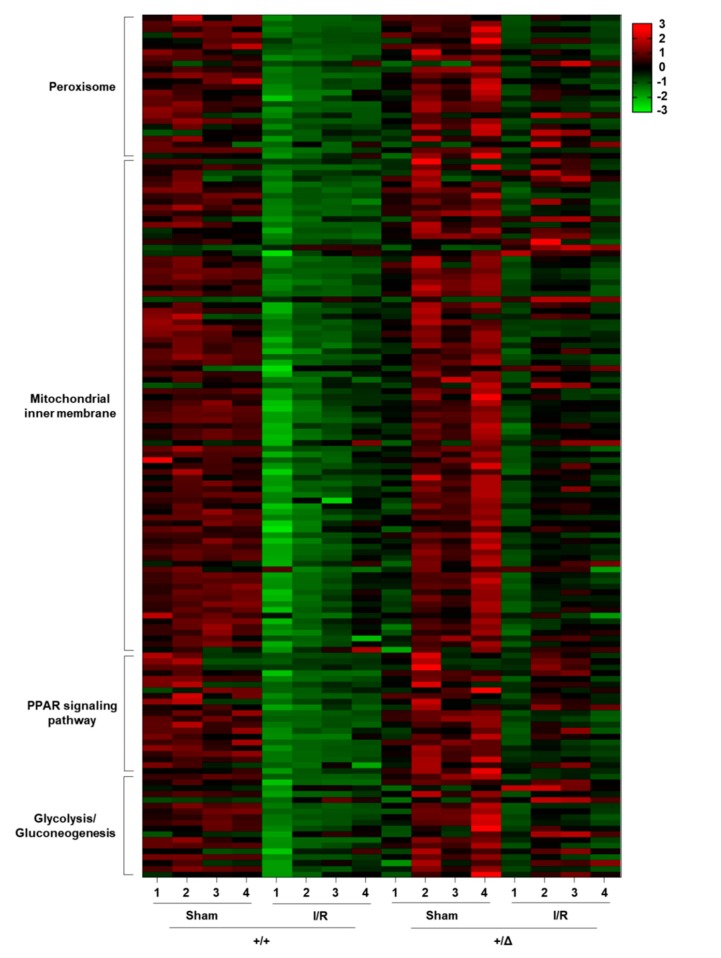
A heatmap of genes whose expression was significantly higher in the *TonEBP^+/Δ^* mice over their *TonEBP^+/+^* littermates after I/R treatment of kidneys, as described in Table 2. Numbers 1 to 4 at the bottom denote individual animals (n = 4).

**Figure 9 cells-08-01284-f009:**
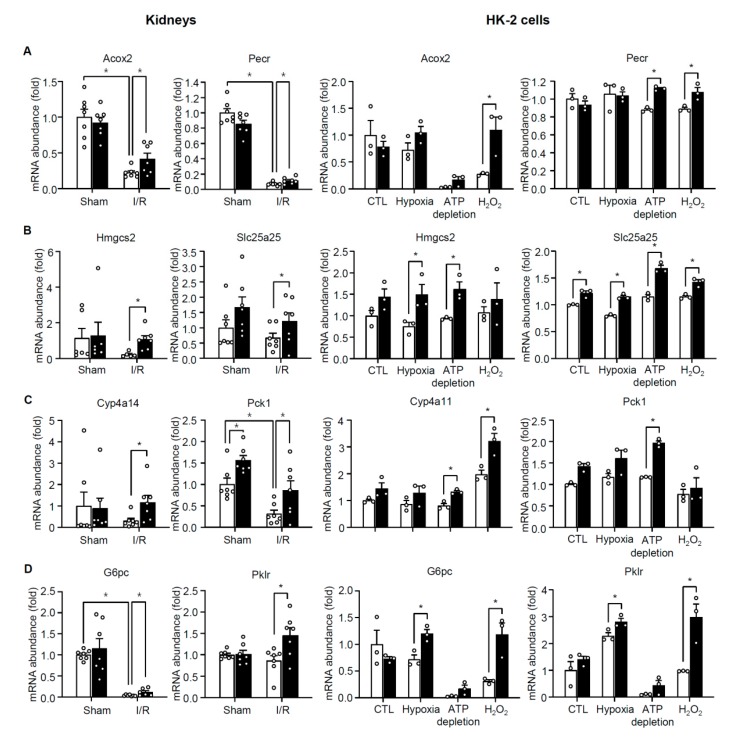
RT-qPCR analyses of kidneys (left) and HK-2 cells (right) for selected genes in the clusters shown in Table 2: (**A**) peroxisome, (**B**) mitochondrial inner membrane, (**C**) PPAR signaling pathways, (**D**) glycolysis/gluconeogenesis. Kidneys were isolated from *TonEBP^+/Δ^* (filled bars) mice and their *TonEBP^+/+^* littermates (open bars) after I/R or sham treatment. HK-2 cells were transfected with scrambled (open bars) or TonEBP-targeted siRNA (filled bars) followed by treatment with hypoxia (1% O_2_), ATP depletion (10 μM of A.A. and 10 mM of 2-DG) or 0.5 mM H_2_O_2_ for 3 h. Mean + SEM, * *p* < 0.05.

**Figure 10 cells-08-01284-f010:**
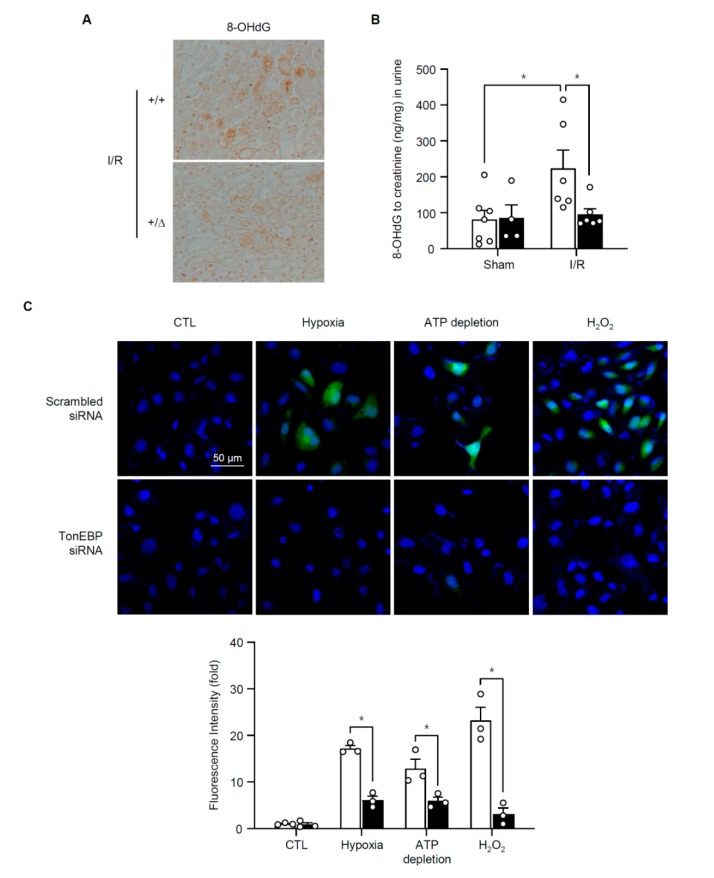
Effects of TonEBP deficiency on oxidative stress in kidneys and HK-2 cells. Renal I/R or sham treatment was performed on *TonEBP^+/Δ^* mice (+/Δ, filled bars) and their *TonEBP^+/+^* littermates (+/+, open bars). (**A**) Kidney sections were stained for 8-OHdG. 8-OHdG and creatinine were measured from urine samples and their ratios in ng/mg are shown (**B**). (**C**) HK-2 cells were transfected with TonEBP targeted (open bars) or scrambled siRNA (filled bars) followed by no treatment (CTL), or treatment with hypoxia, ATP depletion, or H_2_O_2_. Fluorescent images of DCF (green) and Hoechst 33342 (blue) in HK-2 cells loaded with DCF-DA are shown at the top. DCF fluoresence intensity as Mean ± SEM is shown at the bottom. * *p* < 0.05.

**Figure 11 cells-08-01284-f011:**
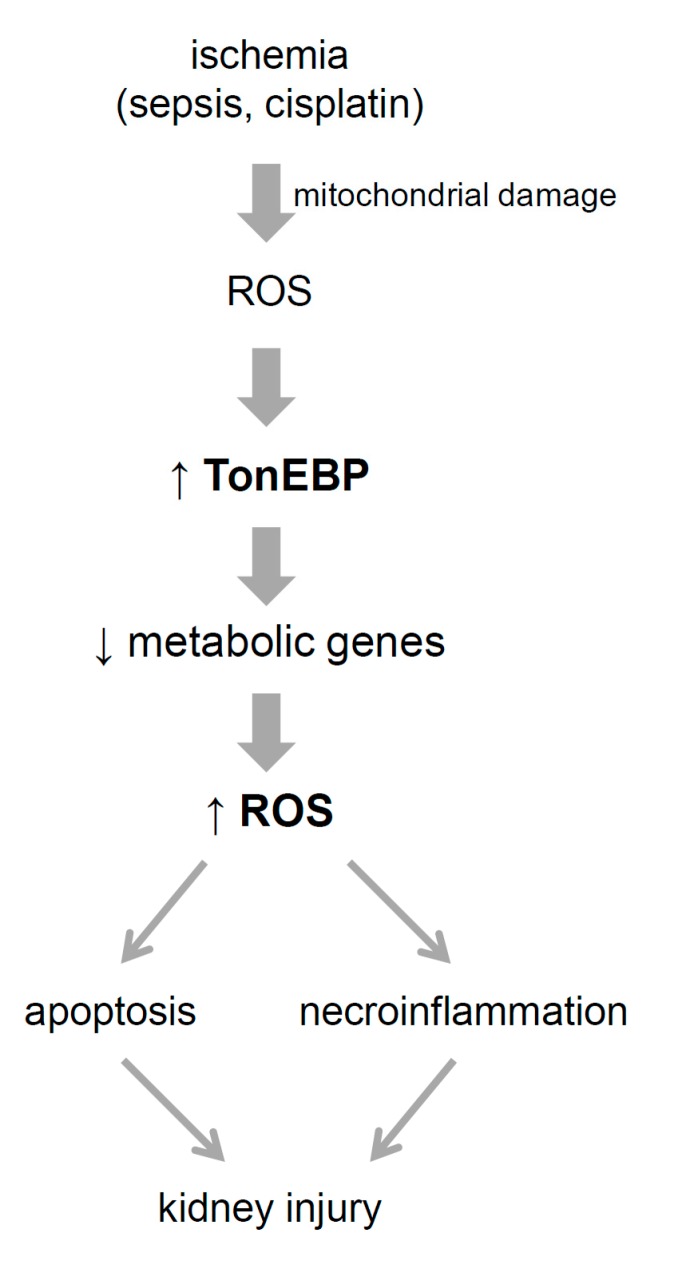
Model for the role of TonEBP in acute kidney injury. Mitochondrial damage in response to ischemia (as well as sepsis and cisplatin) leads to the production of reactive oxygen species (ROS) in addition release of cytochrome c and mitochondrial DNA in renal tubules. Induced by ROS, TonEBP promotes further ROS production via suppression of genes involved in cellular pathways including peroxisome, mitochondrial inner membrane, PPAR signaling, and glycolysis/gluconeogenesis. ROS causes apoptosis and necorinflammation (cell death and inflammation) in tubular cells leading to functional kidney injury. See text for details.

**Table 1 cells-08-01284-t001:** RT-qPCR analyses of inflammatory genes and adhesion molecules in the renal outer medullae of *TonEBP^+/Δ^* mice (+/Δ) and their *TonEBP^+/+^* litter mates (+/+) after I/R or sham treatment of kidneys. Abundance is calculated relative to sham, +/+. Mean ± SEM, n = 6–7. * *p* < 0.05 vs. corresponding +/+. ^#^
*p* < 0.05 vs. corresponding sham.

	Sham	I/R
+/+	+/Δ	+/+	+/Δ
**Cytokines & Chemokines**
IL-6	1.0 ± 0.3	1.5 ± 0.5	29.1 ± 6.5 ^#^	9.3 ± 2.5 ^#,^*
IL-1β	1.0 ± 0.1	1.2 ± 0.2	12.5 ± 1.0 ^#^	9.7 ± 1.3 ^#,^*
IL-18	1.0 ± 0.1	0.9 ± 0.1	1.4 ± 0.2	1.0 ± 0.1 *
TNF-α	1.0 ± 0.1	0.8 ± 1.0 *	5.6 ± 0.5 ^#^	4.6 ± 0.4 ^#^
IFN-γ	1.0 ± 0.2	0.7 ± 0.1	1.7 ± 0.2 ^#^	1.1 ± 0.1 ^#,^*
IP-10	1.0 ± 0.1	1.1 ± 0.2	3.2 ± 0.3 ^#^	2.5 ± 0.1 ^#,^*
RANTES	1.0 ± 0.1	0.7 ± 0.1 *	1.1 ± 0.1	1.2 ± 0.2^#^
IL-10	1.0 ± 0.2	5.3 ± 1.7 *	17.2 ± 2.2 ^#^	9.8 ± 1.8 ^#,^*
MCP-1	1.0 ± 0.1	2.4 ± 0.8 *	40.9 ± 5.1 ^#^	21.9 ± 2.7 ^#,^*
**Adhesion Molecules**
ICAM	1.0 ± 0.1	1.6 ± 0.2 *	1.7 ± 0.1 ^#^	1.4 ± 0.1 *
E-selectin	1.0 ± 0.3	0.9 ± 0.1	3.1± 0.4 ^#^	1.8 ± 0.2 ^#,^*
VCAM-1	1.0 ± 0.1	2.0 ± 0.3 *	3.3 ± 0.4 ^#^	2.8 ± 0.2 ^#^

**Table 2 cells-08-01284-t002:** Pathways of genes whose renal expression is higher in the *TonEBP^+/Δ^* mice compared to their *TonEBP^+/+^* littermates after I/R treatment of kidneys. Microarray analyses were performed on RNA samples (n = 4) obtained from kidneys. Number of Genes denotes (number of genes whose expression was significantly higher in the *TonEBP^+/Δ^* mice)/(total number of genes in that particular pathway). FDR: false discovery rate. See text for details.

Pathway	Number of Genes	FDR q-Value	Selected Genes in Cluster
Peroxisome	25/77	<0.001	Acox2, Pecr, Dao, Hacl1, Amacr, Baat, Hao1, Mpv17l, Decr2, Agxt, Hao2, Abcd4
Mitochondrial Inner Membrane	86/291	<0.001	Hmgc2, Prodh2, Maob, Slc25a25, Slc25a42, Hsd3b3, Gatm, Slc25a10
PPAR Signaling Pathways	21/79	0.0035	Cyp4a14, Hmgcs2, Pck1, Fabp3, Cyp4a32, Scd1, Acox2, Cyp4a10, Apoc3, Gyk
Glycolysis/Gluconeogenesis	18/56	0.0037	G6pc, Pck1, Pklr, Acss2, Pfkm, Aldh3b1, Pgam2, Aldoc, Gatm, Acss1

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
