# Peer review of "Transcriptional Regulator TonEBP Mediates Oxidative Damages in Ischemic Kidney Injury"

_cells, 2019, doi:10.3390/cells8101284_

Round 1

Reviewer 1 Report

The paper by Yoo et al focuses on the role of TonEBP (NFAT5) in an ischemia/reperfusion model. Their TonEBP haplo-deficiency mouse model is by now well known and creates a good platform to study the role of TonEBP in different disease states. In this paper they study the mediating role of TonEBP in conditions that result in cellular stress such as hypoxia, ATP depletion and an H2O2 environment. In general they found that TonEBP appears to worsen the inflammatory response of I/R damage and this may prove to be a therapeutic option in patients with acute kidney injury. 

Although the differences between knocked-out or silenced TonEBP and the wildtype group are usually small, the results are convincing and support the authors' conclusion. All experiments are well controlled for. The paper is well written. 

I have only 2 requests/questions which may preclude publication:
1. Can table 2 be replaced by a heatmap containing all results?
2. Can the authors provide a possible therapeutic in the discussion section which would be possible to use in clinical practice? 

Reviewer 2 Report

Overall, It is an interesting and relevant article. I consider it a useful contribution in its field.

The investigators can consider improve the discussion part to further describe the metabolic genes that are suppressed by increase in TonEBP.

Also, in the introduction, the investigators mentioned "TonEBP is a stress protein that is induced by a variety of stresses including hyperglycemia". Any metabolic genes that got involved for glucose metabolism, and may additionally discuss in the discussion.
